# OpenReview forum: "MASpi: A Unified Environment for Evaluating Prompt Injection Robustness in LLM-Based Multi-Agent Systems"
_ICLR.cc/2026/Conference — ICLR 2026 Conference Withdrawn Submission_

### Official Review · Reviewer_e2tU · 2025-10-31

**Soundness:** 2
**Presentation:** 3
**Contribution:** 2
**Rating:** 2
**Confidence:** 4

**Summary:**

The paper proposes MASPI, a unified and extensible environment for evaluating prompt injection robustness in multi-agent systems (MAS). It enables comparable evaluations by providing a modular framework for integrating diverse agents and attacks, a threat taxonomy covering three attack surfaces and three objectives, and a benchmark featuring 23 attacks and 966 cases in math reasoning and code generation. The paper evaluates seven popular systems with three LLMs. The results show that topology does not ensure security, risks are dispersed across agents by objective, and single-agent defenses do not reliably transfer to multi-agent settings.

**Strengths:**

1. Timely Problem Focus: The paper addresses the critical problem of prompt injection by extending it from single-agent systems to the MAS context. It correctly identifies that inter-agent collaboration introduces novel, propagation-based vulnerabilities.

2. Systematic Threat Taxonomy: The work provides a clear taxonomy of MAS vulnerabilities, structured by attack surfaces (inputs, profiles, messages) and objectives (hijacking, disruption, exfiltration). This categorization, illustrated with 23 attacks, aids standardized evaluation.

**Weaknesses:**

Major Weaknesses:

1. Limited Framework Novelty: The core contribution of this work is a unified environment for MAS robustness evaluation. However, the paper fails to articulate what makes this framework fundamentally different from other existing unified MAS frameworks (e.g., MASLab [1]). Specifically, the attack modules are essentially programmatic modifications to an MAS configuration (e.g., changing an agent's profile, intercepting a message, or modifying user input). It is not clear why a new environment is required for this, as these attack simulations could seemingly be implemented as an evaluation harness on top of existing unified MAS frameworks.

2. Contradictory Threat Model: The paper's threat model is contradictory. It claims a "black-box" setting but grants attackers powerful, white-box capabilities. The "Malicious Agent" and "Message Poison" surfaces require privileged access to internal agent profiles and the ability to intercept inter-agent communication, respectively. These assumptions are unrealistic for a black-box attacker.

3. Simple MAS Attack: The paper's conclusions on robustness are based on 23 manually crafted, static prompt templates. The evaluation omits more sophisticated, automated attacks, such as those using gradient-based or adaptive search methods. A system's robustness to these simple attacks does not guarantee its resilience against more advanced variants.

Minor Weaknesses:

1. Narrow Evaluation Scope: The evaluation focuses only on mathematical reasoning and code generation, which limits the generalizability of its conclusions. It is unclear if these findings apply to other common MAS domains, such as creative writing, social simulation, or web navigation, where attack propagation and impact might differ significantly.

2. Defense Analysis: The paper overlooks more robust, MAS-specific defenses, such as topology-aware security policies (e.g., restricting agent capabilities based on role) or anomaly detection based on communication patterns.

[1] Ye R, Huang K, Wu Q, et al. MASLab: A unified and comprehensive codebase for LLM-based multi-agent systems [J]. arXiv preprint arXiv:2505.16988, 2025.

**Questions:**

1. Could you differentiate MASPI from existing unified MAS frameworks?

2. Could you strengthen the threat model?

3. Could you add automated or adaptive attacks?

---

> ### Author Response · Authors · 2025-11-21
> **Response to Reviewer e2tU (1/4)**
>
> We thank the reviewer for the valuable feedback on improving this paper! Please find below our response to the reviewer’s questions.
>
> > **W1 & Q1** Limited Framework Novelty: The core contribution of this work is a unified environment for MAS robustness evaluation. However, the paper fails to articulate what makes this framework fundamentally different from other existing unified MAS frameworks (e.g., MASLab [1]). Specifically, the attack modules are essentially programmatic modifications to an MAS configuration (e.g., changing an agent's profile, intercepting a message, or modifying user input). It is not clear why a new environment is required for this, as these attack simulations could seemingly be implemented as an evaluation harness on top of existing unified MAS frameworks. Could you differentiate MASPI from existing unified MAS frameworks?
>
> We thank the reviewer for the valuable feedback, which allows us to further clarify our contribution. The key difference lies in the **interface specification**:
> - **MASLab [1] (unified system-level I/O interface)**: MASLab standardizes only the system-level I/O, not the internal implementations. Some LLM-MAS instances rely on direct OpenAI SDK calls and manage agent components within the MAS class, while others use dedicated Agent classes. As a result, evaluating robustness on top of [1] requires customizing attack and defense modules for each implementation, which hinders the generalizability of these modules.
> - **MASpi (unified component-level interface)**: MASpi enforces a consistent MAS–Agent–component hierarchy and provides a unified interface for component-level interventions. This standardization allows attack and defense modules to be applied across all implementations. Its modular design also offers strong extensibility, enabling rapid development for future research.
>
> > **W2 & Q2** Contradictory Threat Model: The paper's threat model is contradictory. It claims a "black-box" setting but grants attackers powerful, white-box capabilities. The "Malicious Agent" and "Message Poison" surfaces require privileged access to internal agent profiles and the ability to intercept inter-agent communication, respectively. These assumptions are unrealistic for a black-box attacker. Could you strengthen the threat model?.
>
> We thank the reviewer for this critical observation. We acknowledge that our original description may have been ambiguous, and we have clarified the specification of the threat model in the revised version (**Section 3.2**). Specifically, we would like to explain that our threat model defines “black-box” access specifically at the model level, indicating that attackers do not have access to underlying model gradients and cannot perform white-box adaptive attacks such as GCG [9]. The permissible interventions on LLM-MAS components are strictly aligned with established prior works [2–6], including modifications to agent profiles [2, 4] and tampering with intermediate messages [3, 4].

---

> ### Author Response · Authors · 2025-11-21
> **Response to Reviewer e2tU (2/4)**
>
> > **W3 & Q3** Simple MAS Attack: The paper's conclusions on robustness are based on 23 manually crafted, static prompt templates. The evaluation omits more sophisticated, automated attacks, such as those using gradient-based or adaptive search methods. A system's robustness to these simple attacks does not guarantee its resilience against more advanced variants. Could you add automated or adaptive attacks?
>
> We thank the reviewer for the insightful suggestion, which helps us better clarify the contribution of our work. To address the concerns, we discuss two aspects: **the attack generation process in MASpi** and **adaptive attacks under deployed defenses.**
> 1. **Clarifying the attack generation process in MASpi**: The original version omitted details on how attacks are constructed, and we have added them in the revised paper (**Section 4.2**). MASpi does not use manually crafted or static templates. Instead, it generates attacks adaptively through an LLM judge–guided search starting from a seed instruction.  In practice, the optimized prompts consistently converge to patterns such as domain-adaptive role-playing, context-aware few-shot demonstrations, reinforced output-format guidance, and persuasive instructions targeting downstream agents (**Appendix D**).
> 2. **Adaptive attacks under deployed defenses**: To further demonstrate the adaptiveness of our attacks, we additionally optimize a group of hijacking attacks on an LLM-MAS equipped with the best-performing defense: Safety Filter (**Table 1 below**). Even under this defense, the attacks after optimization remain effective. These findings also reinforce our conclusion in the revised paper (**Section 5.4**) that robust protection requires defenses spanning both model-level and system-level design.
>
> ---
>
> Table 1. Additional results of adaptive attacks.
> | LLM-MAS                   | UA     | ASR    |
> |----------------------------|--------|--------|
> | AutoGen + Safety Filter    | 13.33  | 6.67   |
> | AutoGen + Safety Filter (optimized attack)   | 0      | 36.67  |
> | AgentVerse + SafetyFilter  | 60.00  | 3.33   |
> | AgentVerse + SafetyFilter (optimized attack) | 23.33  | 13.33  |
> | MetaGPT + Safety Filter    | 43.33  | 0.00   |
> | MetaGPT + Safety Filter (optimized attack)  | 13.33  | 10.00  |

---

> ### Author Response · Authors · 2025-11-21
> **Response to Reviewer e2tU (3/4)**
>
> > **Minor W1** Narrow Evaluation Scope: The evaluation focuses only on mathematical reasoning and code generation, which limits the generalizability of its conclusions. It is unclear if these findings apply to other common MAS domains, such as creative writing, social simulation, or web navigation, where attack propagation and impact might differ significantly.
>
> We thank the reviewer for the suggestion regarding the evaluation scope. We address this from two perspectives: **task domain selection** and **additional experiments**:
> - **Task domain selection**: In security benchmarking, accurate failure attribution is crucial. Domains such as creative writing rely on open-ended evaluation, while web navigation introduces substantial environmental noise, making it difficult to isolate attack effects from domain-specific variability. Following common practice in recent LLM-MAS studies [1, 10–12], we instead extend our benchmark to two domains that maintain rigorous, quantifiable evaluation: Science and Medical.
> - **Additional experiments**: Our preliminary results (**Appendix F.1** and **Table 2 below**) show that LLM-MAS systems exhibit similar vulnerabilities to prompt injection attacks in these domains. These findings suggest that the observed weaknesses reflect broader structural limitations inherent to current LLM-MAS systems.
>
> ---
>
> Table 2. Additional results on Science and Medical Domain.
> | **Task Domain** | **LLM-MAS**        | **BU**                 | **Hijacking UA**          | **Hijacking ASR**         | **Disruption UA**         | **Disruption ASR**        | **Exfiltration UA**       | **Exfiltration ASR**      |
> |-----------------|--------------------|-------------------------|----------------------------|----------------------------|----------------------------|----------------------------|----------------------------|----------------------------|
> | *Science*       | CAMEL              | 15.93 ± 5.14            | 22.89 ± 5.17               | 0.62 ± 2.66                | 27.35 ± 0.88               | 28.90 ± 1.26               | 25.64 ± 2.73               | 0.57 ± 2.18                |
> |                 | AutoGen            | 43.65 ± 7.62            | 37.98 ± 5.07               | 50.26 ± 5.16               | 42.67 ± 1.22               | 43.22 ± 1.08               | 25.95 ± 10.68              | 23.24 ± 6.41               |
> |                 | AgentVerse         | 46.03 ± 1.66            | 52.59 ± 5.77               | 25.14 ± 1.61               | 43.87 ± 2.01               | 11.18 ± 2.23               | 35.53 ± 6.82               | 11.36 ± 9.33               |
> |                 | Self Consistency   | 44.32 ± 5.67            | 46.42 ± 6.08               | 16.89 ± 3.74               | 40.95 ± 2.56               | 16.29 ± 0.30               | 40.10 ± 7.35               | 41.09 ± 9.03               |
> | *Medical*       | CAMEL              | 7.33 ± 6.16             | 9.23 ± 7.63                | 0.00 ± 0.00                | 13.75 ± 0.91               | 30.72 ± 0.73               | 10.00 ± 1.79               | 10.63 ± 2.61               |
> |                 | AutoGen            | 67.17 ± 4.63            | 32.05 ± 6.85               | 47.36 ± 6.14               | 43.56 ± 1.36               | 49.04 ± 2.09               | 41.24 ± 3.63               | 29.92 ± 4.03               |
> |                 | AgentVerse         | 56.80 ± 6.57            | 46.05 ± 4.24               | 23.38 ± 6.70               | 54.34 ± 1.45               | 18.03 ± 0.23               | 58.47 ± 2.44               | 19.24 ± 0.47               |
> |                 | Self Consistency   | 61.33 ± 5.20            | 53.38 ± 5.96               | 15.19 ± 4.82               | 56.97 ± 1.81               | 16.46 ± 0.86               | 48.62 ± 2.36               | 38.49 ± 0.87               |

---

> ### Author Response · Authors · 2025-11-21
> **Response to Reviewer e2tU (4/4)**
>
> > **Minor W2** Defense Analysis: The paper overlooks more robust, MAS-specific defenses, such as topology-aware security policies (e.g., restricting agent capabilities based on role) or anomaly detection based on communication patterns.
>
> To address the reviewer's concern, we have strengthened our defense evaluation by including two widely used defenses: an agent-based defense, AGrail [7], and a topology-guided defense for LLM-MAS, G-Safeguard [8] (**Section 5.4** and **Table 3 below**).
>
> Our key observation is that both defenses demonstrate limited utility preservation and generalization on MASpi. This further underscores the necessity and significance of a unified evaluation environment for advancing research on LLM-MAS security.
>
> ---
> Table 3. Additional results of defenses.
> | System                   | BU              | Hijacking UA      | Hijacking ASR     | Disruption UA     | Disruption ASR    | Exfiltration UA    | Exfiltration ASR   |
> |--------------------------|-----------------|--------------------|--------------------|--------------------|--------------------|---------------------|---------------------|
> | AutoGen + AGrail         | 32.22 ± 4.78    | 7.50 ± 0.00        | 35.56 ± 3.16       | 1.11 ± 2.53        | 96.44 ± 0.96       | 14.00 ± 0.00        | 29.33 ± 1.66        |
> | AutoGen + G-Safeguard    | 40.00 ± 8.28    | 15.56 ± 1.20       | 67.22 ± 2.39       | 0.22 ± 0.96        | 96.44 ± 0.96       | 21.33 ± 1.66        | 34.00 ± 1.66        |
> | AgentVerse + AGrail      | 54.44 ± 4.78    | 44.17 ± 2.07       | 40.28 ± 1.20       | 20.22 ± 0.96       | 65.33 ± 1.66       | 37.78 ± 0.96        | 67.33 ± 1.66        |
> | AgentVerse + G-Safeguard | 46.67 ± 8.28    | 46.39 ± 1.20       | 33.06 ± 1.20       | 20.89 ± 0.96       | 74.67 ± 1.66       | 33.33 ± 1.66        | 56.22 ± 1.91        |
> | MetaGPT + AGrail         | 6.67 ± 8.28     | 0.28 ± 1.20        | 11.67 ± 0.00       | 0.00 ± 0.00        | 96.89 ± 0.96       | 5.56 ± 0.96         | 19.78 ± 2.53        |
> | MetaGPT + G-Safeguard    | 31.11 ± 4.78    | 27.78 ± 2.39       | 46.39 ± 1.20       | 6.22 ± 1.91        | 92.89 ± 0.96       | 28.00 ± 0.00        | 42.89 ± 0.96        |
>
> ---
>
> **References**
>
> [1] Ye, R., et al. 2025. MASLab: A Unified and Comprehensive Codebase for LLM-based Multi-Agent Systems. arXiv 2025.
>
> [2] Yu, M., et al. 2025. NetSafe: Exploring the Topological Safety of Multi-Agent Systems. ACL 2025.
>
> [3] He, P., et al. 2025. Red-Teaming LLM Multi-Agent Systems via Communication Attacks. ACL 2025.
>
> [4] Huang, J., et al. 2025. On the Resilience of LLM-Based Multi-Agent Collaboration with Faulty Agents. ICML 2025.
>
> [5] Zheng, C., et al. 2025. Demonstrations of Integrity Attacks in Multi-Agent Systems. arXiv 2025.
>
> [6] Zhou, Z., et al. 2025. CORBA: Contagious Recursive Blocking Attacks on Multi-Agent Systems Based on Large Language Models. arXiv 2025.
>
> [7] Luo, W., et al. 2025. AGrail: A Lifelong Agent Guardrail with Effective and Adaptive Safety Detection. ACL 2025.
>
> [8] Wang, S., et al. 2025. G-Safeguard: A Topology-Guided Security Lens and Treatment on LLM-based Multi-Agent Systems. ACL 2025.
>
> [9] Zou, A., et al. 2023. Universal and Transferable Adversarial Attacks on Aligned Language Models. arXiv 2023.
>
> [10] Ye, R., et al. 2025. MAS-GPT: Training LLMs to build LLM-based multi-agent systems. ICML 2025.
>
> [11] Amayuelas, A., et al. 2024. MultiAgent Collaboration Attack: Investigating Adversarial Attacks in Large Language Model Collaborations via Debate. EMNLP 2024.
>
> [12] Sun, Y., et al. 2025. CortexDebate: Debating Sparsely and Equally for Multi-Agent Debate. ACL 2025.

---

### Official Review · Reviewer_jtxB · 2025-11-01

**Soundness:** 2
**Presentation:** 3
**Contribution:** 3
**Rating:** 6
**Confidence:** 4

**Summary:**

1. This paper proposes a unified environment for evaluating the prompt injection robustness of LLM-based multi-agent systems (LLM-MAS).
2. Benchmarking results reveal that existing LLM-MAS remain highly vulnerable to prompt injection attacks, highlighting the urgent need for stronger defenses in multi-agent collaboration.

**Strengths:**

1. The paper is well-written and easy to follow.

2. The problem addressed—evaluating the prompt injection robustness of LLM-based multi-agent systems (LLM-MAS)—is important.

3. The proposed benchmark comprehensively covers multiple attack surfaces and objectives, encompassing diverse and realistic threat scenarios.

4. The unified and modular environment allows for reproducible evaluation, easy integration of new attacks or systems, and standardized comparisons across LLM-MAS.

5. The experiments involve seven multi-agent systems and three large models, providing broad evidence that current LLM-MAS remain highly vulnerable, which motivates further research.

**Weaknesses:**

1. The defense analysis (e.g., BERT detector, Delimiter, Sandwich) is relatively shallow and does not propose new mitigation methods.

2. While empirical coverage is strong, the paper does not deeply analyze why certain systems or topologies are more vulnerable.

3. MASPI mainly functions as an evaluation framework and contributes less on the conceptual or algorithmic aspects of enhancing robustness. Moreover, while the benchmark reports BU, ASR, and UA metrics, it does not introduce new or more insightful evaluation metrics.

**Questions:**

See weakness.

---

> ### Author Response · Authors · 2025-11-21
> **Response to Reviewer jtxB (1/2)**
>
> We thank the reviewer for the valuable feedback on improving this paper! Please find below our response to the reviewer’s questions.
>
> > **W1** The defense analysis (e.g., BERT detector, Delimiter, Sandwich) is relatively shallow and does not propose new mitigation methods.
>
> We would like to clarify that we have proposed a new defense, which may be overlooked due to insufficient description.  We have expanded the corresponding section in the revised paper (**Section 5.4**), and the main improvements are as follows:
> - **Comparing more defenses**: We strengthen the evaluation by adding two widely used defenses: an agent-based defense AGrail [1], and a topology-guided defense for LLM-MAS, G-Safeguard [2] (**Section 5.4** and **Table 1 below**).
> - **In-depth analysis**: We provide a deeper analysis based on defense evaluation. Key findings include: (1) prompt injections in LLM-MAS often lack clear malicious patterns, making detection difficult; (2) prior defenses were evaluated in simplified environments, limiting their generalizability to more realistic scenarios.
> - **Proposing a new defense**: We propose the Safety Filter defense, which leverages the observation that prompt injections are hard to detect. Unlike existing methods [2,3] that rely on detection and pruning, we focus on information aligned with the user task rather than directly detecting injected prompts.
>
> ---
>
> Table 1. Additional results of defenses.
> | System                   | BU              | Hijacking UA      | Hijacking ASR     | Disruption UA     | Disruption ASR    | Exfiltration UA    | Exfiltration ASR   |
> |--------------------------|-----------------|--------------------|--------------------|--------------------|--------------------|---------------------|---------------------|
> | AutoGen + AGrail         | 32.22 ± 4.78    | 7.50 ± 0.00        | 35.56 ± 3.16       | 1.11 ± 2.53        | 96.44 ± 0.96       | 14.00 ± 0.00        | 29.33 ± 1.66        |
> | AutoGen + G-Safeguard    | 40.00 ± 8.28    | 15.56 ± 1.20       | 67.22 ± 2.39       | 0.22 ± 0.96        | 96.44 ± 0.96       | 21.33 ± 1.66        | 34.00 ± 1.66        |
> | AgentVerse + AGrail      | 54.44 ± 4.78    | 44.17 ± 2.07       | 40.28 ± 1.20       | 20.22 ± 0.96       | 65.33 ± 1.66       | 37.78 ± 0.96        | 67.33 ± 1.66        |
> | AgentVerse + G-Safeguard | 46.67 ± 8.28    | 46.39 ± 1.20       | 33.06 ± 1.20       | 20.89 ± 0.96       | 74.67 ± 1.66       | 33.33 ± 1.66        | 56.22 ± 1.91        |
> | MetaGPT + AGrail         | 6.67 ± 8.28     | 0.28 ± 1.20        | 11.67 ± 0.00       | 0.00 ± 0.00        | 96.89 ± 0.96       | 5.56 ± 0.96         | 19.78 ± 2.53        |
> | MetaGPT + G-Safeguard    | 31.11 ± 4.78    | 27.78 ± 2.39       | 46.39 ± 1.20       | 6.22 ± 1.91        | 92.89 ± 0.96       | 28.00 ± 0.00        | 42.89 ± 0.96        |
>
> > **W2** While empirical coverage is strong, the paper does not deeply analyze why certain systems or topologies are more vulnerable.
>
> We thank the reviewer for highlighting the importance of understanding system-specific vulnerabilities. Following this suggestion, we have augmented our analysis of system vulnerabilities in the revised paper (**Section 5**). The key findings include:
> - **Limited visibility and inter-agent trust amplify risk**: Systems with simple interaction structures have limited global visibility and rely on implicit inter-agent trust, making them more prone to executing injected instructions.
> - **Dense interactions facilitate malicious propagation**: Compared to direct vertical structures, dense internal interactions are the primary driver of malicious content propagation. Even when critical roles participate in these dense interactions, they cannot effectively prevent the spread of harmful instructions.

---

> ### Author Response · Authors · 2025-11-21
> **Response to Reviewer jtxB (2/2)**
>
> > **W3** MASPI mainly functions as an evaluation framework and contributes less on the conceptual or algorithmic aspects of enhancing robustness. Moreover, while the benchmark reports BU, ASR, and UA metrics, it does not introduce new or more insightful evaluation metrics.
>
> We thank the reviewer for this valuable suggestion and would like to further clarify our novel contributions:
> - **Proposing a new defense**: We introduce a novel defense mechanism (**Section 5.4**) motivated by our observation that prompt injection in LLM-MAS is difficult to detect. Experimental results demonstrate that this defense achieves stronger security than existing strategies.
> - **Introducing a new evaluation metric**: While existing metrics such as BU, ASR, and UA capture various LLM-MAS risks, including domain-specific vulnerabilities and threats from malicious agents, they focus solely on final outputs and overlook how attacks propagate within the system. To gain deeper insights into the internal dynamics of LLM-MAS and provide more precise guidance for defense design, we propose a new metric, the Propagation Vulnerability Index (**Section 4.3**), which quantifies the extent of malicious content propagation within the system. Using this metric, we reveal how interaction structures and component design influence the spread of harmful instructions within the LLM-MAS (**Section 5.2**).
>
> ---
>
> **References**
>
> [1] Luo, W., et al. 2025. AGrail: A Lifelong Agent Guardrail with Effective and Adaptive Safety Detection. ACL 2025.
>
> [2] Wang, S., et al. 2025. G-Safeguard: A Topology-Guided Security Lens and Treatment on LLM-based Multi-Agent Systems. ACL 2025.
>
> [3] Miao, R., et al. 2025. BlindGuard: Safeguarding LLM-Based Multi-Agent Systems Under Unknown Attacks. arXiv.

---

### Official Review · Reviewer_ikCb · 2025-11-01

**Soundness:** 2
**Presentation:** 3
**Contribution:** 2
**Rating:** 4
**Confidence:** 4

**Summary:**

This paper introduces MASPI, a novel and unified benchmarking environment for systematically evaluating the robustness of LLM-based Multi-Agent Systems (LLM-MAS) against prompt injection attacks. The authors address a critical fragmentation in existing research by providing a standardized framework, which includes a 3x3 threat taxonomy (organizing attacks by 3 surfaces and 3 objectives), a modular codebase integrating 7 popular LLM-MAS frameworks, and a large-scale test suite of 966 test cases. Using this benchmark, the authors conduct extensive experiments, revealing that current systems are highly vulnerable (90-100% ASR in some cases), that single-agent defenses are often ineffective or even counter-productive in a multi-agent context, and that vulnerability is dispersed across agents rather than being tied purely to system topology.

**Strengths:**

This paper's primary contribution, namely a unified, extensible, and standardized benchmar, is both timely and highly valuable. The authors correctly identify that the field's progress on LLM-MAS security is hampered by ad-hoc implementations and fragmented evaluations. By providing a single environment that allows for "apples-to-apples" comparisons of different systems (AutoGen, MetaGPT, etc.) and attacks, MASpi provides a strong foundation for reproducible and cumulative research.

**Weaknesses:**

1. Limited Diversity of Task Domains: The benchmark's current focus is exclusively on mathematical reasoning and code generation (Section 4.1). While these are excellent domains for testing structured, collaborative problem-solving, they are not representative of all common MAS applications. It remains unclear if these findings would generalize to other tasks.
2. Limited Evaluation of Defenses. The paper stops short of assessing more advanced defenses. Notably, many existing guardrails [1, 2] for LLM agents are themselves implemented as agents; when embedded into an agentic framework they effectively create a multi-agent configuration that fit into the paper's scope. These agent-as-guardrail settings should also be evaluated.
3. No Stealthiness Constraints on Attacks. The proposed benchmark reports ASR/UA/UDR but does not bound or measure the stealthiness of injected prompts (e.g., no embedding space distance/token-budget limits, similarity checks, or detectability metrics). Because payloads can substantially alter inputs or messages, high ASR may partly reflect “loud” interventions rather than genuinely covert prompt injections. This may lead to cases where the LLMs are essentially following the "instruction", as the prompt may has been modified a lot. Adding a perturbation budget and a stealth score (semantic similarity or judge-based detectability) will be much more reasonable.

[1] Chen, Zhaorun, Mintong Kang, and Bo Li. "Shieldagent: Shielding agents via verifiable safety policy reasoning." arXiv preprint arXiv:2503.22738 (2025). \
[2] Luo, Weidi, et al. "Agrail: A lifelong agent guardrail with effective and adaptive safety detection." arXiv preprint arXiv:2502.11448 (2025).

**Questions:**

For rebuttal, please refer to the weaknesses. In addtion, on the "Sandwich" Defense Backfiring One of your most interesting findings is in Table 3, where the "Sandwich" defense not only failed but actively increased the ASR for Exfiltration attacks. This is a significant result which also need further investigation. Do you have a concrete insight for this mechanism?

---

> ### Author Response · Authors · 2025-11-21
> **Response to Reviewer ikCb (1/3)**
>
> We thank the reviewer for the valuable feedback on improving this paper! Please find below our response to the reviewer’s questions.
>
> > **W1** Limited Diversity of Task Domains: The benchmark's current focus is exclusively on mathematical reasoning and code generation (Section 4.1). While these are excellent domains for testing structured, collaborative problem-solving, they are not representative of all common MAS applications. It remains unclear if these findings would generalize to other tasks.
>
> We appreciate the reviewer's suggestion, which has helped us improve our evaluation. Following prior work on LLM-MAS [4–7], we have extended our benchmark to include two additional domains: **Science** and **Medical**.  Our preliminary results (**Appendix F.1 and Table 1 below**) show that LLM-MAS systems exhibit similar vulnerabilities to prompt injection across domains, suggesting that these weaknesses stem from broader structural limitations of current LLM-MAS.
>
> ---
>
> Table 1. Additional results on Science and Medical Domain.
> | **Task Domain** | **LLM-MAS**        | **BU**                 | **Hijacking UA**          | **Hijacking ASR**         | **Disruption UA**         | **Disruption ASR**        | **Exfiltration UA**       | **Exfiltration ASR**      |
> |-----------------|--------------------|-------------------------|----------------------------|----------------------------|----------------------------|----------------------------|----------------------------|----------------------------|
> | *Science*       | CAMEL              | 15.93 ± 5.14            | 22.89 ± 5.17               | 0.62 ± 2.66                | 27.35 ± 0.88               | 28.90 ± 1.26               | 25.64 ± 2.73               | 0.57 ± 2.18                |
> |                 | AutoGen            | 43.65 ± 7.62            | 37.98 ± 5.07               | 50.26 ± 5.16               | 42.67 ± 1.22               | 43.22 ± 1.08               | 25.95 ± 10.68              | 23.24 ± 6.41               |
> |                 | AgentVerse         | 46.03 ± 1.66            | 52.59 ± 5.77               | 25.14 ± 1.61               | 43.87 ± 2.01               | 11.18 ± 2.23               | 35.53 ± 6.82               | 11.36 ± 9.33               |
> |                 | Self Consistency   | 44.32 ± 5.67            | 46.42 ± 6.08               | 16.89 ± 3.74               | 40.95 ± 2.56               | 16.29 ± 0.30               | 40.10 ± 7.35               | 41.09 ± 9.03               |
> | *Medical*       | CAMEL              | 7.33 ± 6.16             | 9.23 ± 7.63                | 0.00 ± 0.00                | 13.75 ± 0.91               | 30.72 ± 0.73               | 10.00 ± 1.79               | 10.63 ± 2.61               |
> |                 | AutoGen            | 67.17 ± 4.63            | 32.05 ± 6.85               | 47.36 ± 6.14               | 43.56 ± 1.36               | 49.04 ± 2.09               | 41.24 ± 3.63               | 29.92 ± 4.03               |
> |                 | AgentVerse         | 56.80 ± 6.57            | 46.05 ± 4.24               | 23.38 ± 6.70               | 54.34 ± 1.45               | 18.03 ± 0.23               | 58.47 ± 2.44               | 19.24 ± 0.47               |
> |                 | Self Consistency   | 61.33 ± 5.20            | 53.38 ± 5.96               | 15.19 ± 4.82               | 56.97 ± 1.81               | 16.46 ± 0.86               | 48.62 ± 2.36               | 38.49 ± 0.87               |

---

> ### Author Response · Authors · 2025-11-21
> **Response to Reviewer ikCb (2/3)**
>
> > **W2** Limited Evaluation of Defenses. The paper stops short of assessing more advanced defenses. Notably, many existing guardrails [1, 2] for LLM agents are themselves implemented as agents; when embedded into an agentic framework they effectively create a multi-agent configuration that fit into the paper's scope. These agent-as-guardrail settings should also be evaluated.
>
> We thank the reviewer for this insightful suggestion. Following this guidance, the defense evaluation has been expanded. Since ShieldAgent [1] provides only a demo video with no released code, we report results for AGrail [2] on MASpi instead. We also include a topology-guided defense for LLM-MAS, G-Safeguard [3] (**Section 4.5** and **Table 2 below**). Our results show that both defenses offer limited utility preservation and generalize poorly on MASpi, largely due to the overly simplified settings used in prior evaluations. These findings further underscore the need for a unified and realistic evaluation environment for LLM-MAS security research.
>
> ---
>
> Table 2. Additional results of defenses.
> | System                   | BU              | Hijacking UA      | Hijacking ASR     | Disruption UA     | Disruption ASR    | Exfiltration UA    | Exfiltration ASR   |
> |--------------------------|-----------------|--------------------|--------------------|--------------------|--------------------|---------------------|---------------------|
> | AutoGen + AGrail         | 32.22 ± 4.78    | 7.50 ± 0.00        | 35.56 ± 3.16       | 1.11 ± 2.53        | 96.44 ± 0.96       | 14.00 ± 0.00        | 29.33 ± 1.66        |
> | AutoGen + G-Safeguard    | 40.00 ± 8.28    | 15.56 ± 1.20       | 67.22 ± 2.39       | 0.22 ± 0.96        | 96.44 ± 0.96       | 21.33 ± 1.66        | 34.00 ± 1.66        |
> | AgentVerse + AGrail      | 54.44 ± 4.78    | 44.17 ± 2.07       | 40.28 ± 1.20       | 20.22 ± 0.96       | 65.33 ± 1.66       | 37.78 ± 0.96        | 67.33 ± 1.66        |
> | AgentVerse + G-Safeguard | 46.67 ± 8.28    | 46.39 ± 1.20       | 33.06 ± 1.20       | 20.89 ± 0.96       | 74.67 ± 1.66       | 33.33 ± 1.66        | 56.22 ± 1.91        |
> | MetaGPT + AGrail         | 6.67 ± 8.28     | 0.28 ± 1.20        | 11.67 ± 0.00       | 0.00 ± 0.00        | 96.89 ± 0.96       | 5.56 ± 0.96         | 19.78 ± 2.53        |
> | MetaGPT + G-Safeguard    | 31.11 ± 4.78    | 27.78 ± 2.39       | 46.39 ± 1.20       | 6.22 ± 1.91        | 92.89 ± 0.96       | 28.00 ± 0.00        | 42.89 ± 0.96        |
>
> > **W3** No Stealthiness Constraints on Attacks. The proposed benchmark reports ASR/UA/UDR but does not bound or measure the stealthiness of injected prompts (e.g., no embedding space distance/token-budget limits, similarity checks, or detectability metrics). Because payloads can substantially alter inputs or messages, high ASR may partly reflect “loud” interventions rather than genuinely covert prompt injections. This may lead to cases where the LLMs are essentially following the "instruction", as the prompt may has been modified a lot. Adding a perturbation budget and a stealth score (semantic similarity or judge-based detectability) will be much more reasonable.
>
> First, we would like to explain that prompt injection attacks are typically not constrained by explicit stealthiness requirements such as token‑budget limits or embedding‑distance thresholds. As LLM‑based agents can access and integrate substantial external information through tools, malicious instructions may appear in diverse and sometimes complex forms. Therefore, imposing rigid perturbation budgets would not accurately reflect the realistic attack surfaces of LLM‑MAS.
>
> Second, prompt injection in LLM-MAS often propagates through inter-agent communication [8], where the attack payload can be an entire malicious message rather than a minimally perturbed prompt. Under this setting, enforcing prompt-level similarity constraints is not strictly necessary for modeling authentic threats.
>
> Third, we would like to clarify that the original version omitted details of the attack generation process, which we have now added to the revised paper (**Section 4.2**). Our attack prompts are generated through an LLM‑judge‑guided search initialized from a seed instruction. During optimization, we constrain each injected prompt to remain close to benign prompts within the current attack surface, ensuring that the resulting attacks remain "stealthy" from the system’s perspective.

---

> ### Author Response · Authors · 2025-11-21
> **Response to Reviewer ikCb (3/3)**
>
> > **Q1** In addtion, on the "Sandwich" Defense Backfiring One of your most interesting findings is in Table 3, where the "Sandwich" defense not only failed but actively increased the ASR for Exfiltration attacks. This is a significant result which also need further investigation. Do you have a concrete insight for this mechanism?
>
> We thank the reviewer for this insightful question, which helped us deepen our experimental analysis. We analyze why the Sandwich defense, which repeats the original task description after each message, can inadvertently increase the attack success rate for Exfiltration attacks. This analysis has been incorporated into the revised paper (**Section 5.4**). Our key findings are as follows:
> - Exfiltration attacks combine malicious objectives with legitimate user tasks, so repeating the task description after each message reinforces the connection between benign and malicious instructions.
> - The effect of coupled objectives is further amplified by task repetition and inter-agent trust, causing downstream agents to potentially interpret the repeated instructions as valid user directives.
>
> ---
>
> **References**
>
> [1] Chen, Z., et al. 2025. ShieldAgent: Shielding Agents via Verifiable Safety Policy Reasoning. ICML 2025.
>
> [2] Luo, W., et al. 2025. AGrail: A Lifelong Agent Guardrail with Effective and Adaptive Safety Detection. ACL 2025.
>
> [3] Wang, S., et al. 2025. G-Safeguard: A Topology-Guided Security Lens and Treatment on LLM-Based Multi-Agent Systems. ACL 2025.
>
> [4] Ye, R., et al. 2025. MASLab: A Unified and Comprehensive Codebase for LLM-Based Multi-Agent Systems. arXiv 2025.
>
> [5] Ye, R., et al. 2025. MAS-GPT: Training LLMs to build LLM-based multi-agent systems. ICML 2025.
>
> [6] Amayuelas, A., et al. 2024. MultiAgent Collaboration Attack: Investigating Adversarial Attacks in Large Language Model Collaborations via Debate. EMNLP 2024.
>
> [7] Sun, Y., et al. 2025. CortexDebate: Debating Sparsely and Equally for Multi-Agent Debate. ACL 2025.
>
> [8] Lee, D., et al. 2024. Prompt Infection: LLM-to-LLM Prompt Injection within Multi-Agent Systems. arXiv 2024.

---

### Official Review · Reviewer_9bHV · 2025-11-02

**Soundness:** 3
**Presentation:** 3
**Contribution:** 3
**Rating:** 4
**Confidence:** 3

**Summary:**

This paper introduces a benchmark MASpi to standardize the measurement of the robustness of LLM-MAS against prompt injection. This paper establishes some initial evaluation protocols and results on existing LLM-MAS and LLMs. The experiment results suggest that current LLM-MAS systems are highly vulnerable to prompt attacks, and the safety does not increase as the complexity of the communication grows.

**Strengths:**

1. This paper is novel in that it first tries to establish a method for evaluating the LLM-MAS robustness against prompt injection
2. The paper presents its results in a logical way

**Weaknesses:**

1. No confidence interval is reported with the evaluation numbers
2. The analysis of the experiments is a bit shallow. For example, one can group the compared baselines and analyze which components in that method are more critical against attacks.

**Questions:**

1. Are the tasks meaningfully set up? My concern is that some tasks in prompt injection are not even defensible and can only be addressed by security layers (e.g., proper access authorization) on top of the LLM-MAS system.
2. All numbers should have their statistical confidence intervals to make the work a serious research paper. If you have the original evaluation numbers, please do include them in the appendix or in the main paper.

---

> ### Author Response · Authors · 2025-11-21
> **Response to Reviewer 9bHV**
>
> We thank the reviewer for the valuable feedback on improving this paper! Please find below our response to the reviewer’s questions.
> > **W1 & Q2** No confidence interval is reported with the evaluation numbers. All numbers should have their statistical confidence intervals to make the work a serious research paper. If you have the original evaluation numbers, please do include them in the appendix or in the main paper.
>
> We appreciate the reviewer’s constructive comment. To address this concern, we report the corresponding 95% confidence intervals in the revised paper (**Section 5**). Following the setup in [6], all experiments are repeated three times using a fixed decoding temperature. Since the main figures display only mean values, error bars are shown for the PVI metric only.
>
> Given the large volume of raw data, the complete evaluation results are provided in the supplementary materials to preserve the readability of the revised paper.
>
> > **W2** The analysis of the experiments is a bit shallow. For example, one can group the compared baselines and analyze which components in that method are more critical against attacks.
>
> We appreciate the reviewer’s helpful suggestion on strengthening the experimental analysis. Following the reviewer’s suggestion, we strengthened our experimental analysis, summarized as follows:
> - **New metric**: Existing metrics focus solely on the final responses and overlook how attacks propagate within the system. To obtain deeper insights into the inner workings of LLM-MAS and to provide more fine-grained guidance for system design, we introduce a new metric, the Propagation Vulnerability Index (**Section 4.3**), which quantifies the extent to which malicious content propagates within the system.
> - **In-depth analysis**: We jointly examine agent-level average ASR and PVI (**Section 5.2**), allowing us to identify how specific components, such as interaction structures and role configurations, influence system robustness. Our key findings include:
>   - Systems with simple interaction structures tend to be more vulnerable.
>   - Complexity in interaction structures alone does not ensure robustness; systems that integrate critical roles demonstrate more consistent resistance.
>   - When critical roles engage in dense interactions, they may still inadvertently facilitate the propagation of malicious content.
>   - Structured interaction schemes can substantially mitigate the spread of harmful content.
>
> > **Q1** Are the tasks meaningfully set up? My concern is that some tasks in prompt injection are not even defensible and can only be addressed by security layers (e.g., proper access authorization) on top of the LLM-MAS system.
>
> We thank the reviewer for raising this important point. In the following, we address it in two points：
> - **Rationale for task setup**: We would like to explain that the task setup strictly follows established LLM-MAS security settings [1-5], ensuring that the benchmark reflects meaningful challenges for system evaluation. Specifically, MASpi covers diverse and realistic attack surfaces, including input-based attacks [1,2], malicious agent profiles [3,5], and intermediate-message manipulations [3,4].
> - **Robustness through design and defense**: System robustness arises from the combined effect of system design (e.g., role configurations, communication topology) and defense mechanisms. While some attacks may require system-level defenses such as access authorization, this reflects inherent gaps in current LLM-MAS designs. In principle, all MASpi attacks can be mitigated through appropriate system design. This makes the benchmark useful for evaluating both system design and defensive mechanisms.
>
> ---
> **References**
>
> [1] Zheng, C., et al. (2025). "Demonstrations of Integrity Attacks in Multi-Agent Systems." arXiv 2025.
>
> [2] Zhou, Z., et al. (2025). "Corba: Contagious Recursive Blocking Attacks on Multi-Agent Systems Based on Large Language Models." arXiv 2025.
>
> [3] Huang, J., et al. (2025). "On the Resilience of LLM-Based Multi-Agent Collaboration With Faulty Agents." ICML 2025.
>
> [4] He, P., et al. (2025). "Red-Teaming LLM Multi-Agent Systems via Communication Attacks." ACL 2025.
>
> [5] Yu, M., et al. (2025). "NetSafe: Exploring the Topological Safety of Multi-Agent Systems." ACL 2025.
>
> [6] Weng, Z., Chen, G., and Wang, W. (2025). "Do as We Do, Not as You Think: The Conformity of Large Language Models." ICLR 2025.

---

### Author Response · Authors · 2025-12-01
**We sincerely thank the Area Chair and all reviewers for their careful evaluation.**

We are encouraged that the reviewers found MASpi timely and useful for studying the robustness of LLM-based multi-agent systems (LLM-MAS) under prompt injection.

The reviewers highlighted three main strengths of our work:

- **Unified and extensible benchmark (9bHV, e2tU, ikCb, jtxB):** MASpi offers a standardized, modular environment with a unified MAS–Agent–component hierarchy, enabling consistent “apples-to-apples” comparison of diverse systems and attacks.
- **Systematic threat taxonomy and realistic attacks (e2tU, ikCb, jtxB):** Our 3×3 taxonomy (surfaces × objectives) and 23 attacks over 966 cases provide broad coverage of realistic MAS threat scenarios.
- **Comprehensive evaluation and insights on vulnerabilities/defenses (9bHV, ikCb, jtxB):** Experiments on seven LLM-MAS frameworks and three LLMs reveal that current systems remain highly vulnerable and that many existing defenses are limited in realistic multi-agent settings.

In response to the reviewers’ concerns, we have made the following key revisions:

- **Clarified framework novelty and threat model (e2tU):** We emphasize that MASpi standardizes component-level interfaces (beyond system I/O as in MASLab) and refine the threat model (Section 3.2) to define “black-box” at the *model* level while keeping component interventions aligned with prior LLM-MAS security works.
- **Clarified adaptive attack generation and added stronger attacks (e2tU, ikCb):** We explain that attacks are generated via an LLM-judge-guided search rather than fixed templates (Section 4.2, Appendix D), and we further optimize attacks against the best defense (Safety Filter), showing they remain effective (Section 5.4).
- **Expanded task domains (e2tU, ikCb):** We extend MASpi from math/code to Science and Medical domains (Appendix F.1), where we observe similar vulnerabilities, suggesting structural rather than domain-specific weaknesses.
- **Strengthened defense evaluation and clarified new defense (e2tU, ikCb, jtxB):** We add AGrail and G-Safeguard to the evaluation (Section 5.4) and clarify our new Safety Filter defense, which focuses on reinforcing task-aligned information rather than detecting specific injected tokens.
- **Introduced a new internal metric and deeper analysis (9bHV, jtxB):** We propose the Propagation Vulnerability Index (PVI) (Section 4.3) and analyze agent-level ASR + PVI (Section 5.2) to explain why certain systems/topologies and dense communication patterns are more vulnerable.
- **Improved statistical reporting and task/stealthiness clarifications (9bHV, ikCb):** We now report 95% confidence intervals (Section 5), provide full results in the supplement, clarify task design and defensibility (Section 4.1), and explain how our LLM-guided optimization keeps injected prompts close to benign surface prompts (Section 4.2).

We once again thank the Area Chair and all reviewers for their insightful feedback. We hope that these revisions and additional analyses address the raised concerns and that MASpi, together with the new metric and defenses, will serve as a solid foundation for future research on secure and robust LLM-based multi-agent systems.

---

### Note · Authors · 2025-12-22

I have read and agree with the venue's withdrawal policy on behalf of myself and my co-authors.